# Natural History of Oral HPV Infection among Indigenous South Australians

**DOI:** 10.3390/v15071573

**Published:** 2023-07-18

**Authors:** Xiangqun Ju, Sneha Sethi, Annika Antonsson, Joanne Hedges, Karen Canfell, Megan Smith, Gail Garvey, Richard M. Logan, Lisa M. Jamieson

**Affiliations:** 1Adelaide Dental School, University of Adelaide, Adelaide, SA 5000, Australia; sneha.sethi@adelaide.edu.au (S.S.); joanne.hedges@adelaide.edu.au (J.H.); richard.logan@adelaide.edu.au (R.M.L.); lisa.jamieson@adelaide.edu.au (L.M.J.); 2QIMR Berghofer Medical Research Institute, Brisbane, QLD 4006, Australia; annika.antonsson@qimrberghofer.edu.au; 3Faculty of Medicine, The University of Queensland, Brisbane, QLD 4072, Australia; 4The Daffodil Centre, The University of Sydney, A Joint Venture with Cancer Council NSW, Sydney, NSW 2006, Australiamegan.smith@nswcc.org.au (M.S.); 5School of Public Health, Faculty of Medicine, The University of Queensland, St. Lucia, QLD 4072, Australia; g.garvey@uq.edu.au

**Keywords:** human papilloma viruses (HPV), oropharyngeal cancer, indigenous, incidence, persistence

## Abstract

This study aims to describe the natural history of and identify the risk factors associated with oral human papillomavirus (HPV) infections in an Australian Indigenous cohort. A longitudinal cohort study design, with baseline (2018), 12-month, and 24-month data obtained from Indigenous Australians aged 18+ years in South Australia, was performed. Face-to-face interviews were conducted, and saliva samples for HPV testing were collected at each time point. Basic descriptive analyses were conducted to calculate prevalence, incidence, persistence, clearance, and incidence proportions of any HPV infection. Multivariable logistic regression analyses with adjusted prevalence ratios (PRs) were conducted to identify risk factors associated with oral HPV infection. Among 993 participants with valid saliva samples, 44 HPV types were identified. The prevalence of infection with any oral HPV infection was 51.3%, high-risk HPV was 11%, and types implicated in Heck’s disease (HPV 13 or 32) was 37.4%. The incidence, persistence, and clearance of any and high-risk HPV infections were 30.7%, 11.8% and 33.3% vs. 9.3%, 2.8%, and 9%, respectively. Our findings indicate that the prevalence, incidence, and persistence of oral HPV infection in a large sample of Indigenous Australians were high, and clearance was low. Oral sex behaviours and recreational drug use were risk factors associated with incident high-risk HPV infection.

## 1. Introduction

Human papillomaviruses (HPVs) are the most common, but preventable, sexually transmitted infection in high-income countries [1] including Australia [2,3]. HPVs, which belong to the Papillomaviridae family, consist of a non-enveloped virus with a 50 nm diameter capsid that encloses a single circular double-stranded DNA molecule, approximately 8000 bp in length, associated to cellular histones [4]. HPV types are classified as low-risk (lr) and high-risk (hr) genotypes according to their oncogenic potential. Most males and females who engage in sexual activity will become infected with HPV at least once during their lifetimes. Most HPV infections are transient, undetectable, and do not cause any symptoms and lesions, and viral clearance by the immune system occurs spontaneously within 2 years in more than 90% of infected individuals. Nevertheless, in persistent hr-HPV infections or those that take longer to clear, precancerous lesions are likely to occur [4]. HPV infection is linked with several disease conditions, including anogenital warts [5], Heck’s disease [6,7,8], and cancers (such as cervix, vulva, penis, anus, and oropharyngeal (such as tonsillar and base of tongue) cancers) [9,10].

The incidence of HPV-related oropharyngeal cancers (OPCs) has been increasing in some high-income countries and, in some settings (with longstanding and effective cervical screening), overtaking cervical cancer [11,12]. There were 92,887 new cases of OPCs and 51,005 deaths due to OPCs worldwide in 2018 [13]. OPCs have been associated with oral hr-HPV infection and are usually of type 16 [2]. Other high-risk types include 18, 26, 31, 33, 35, 39, 45, 51, 52, 53, 56, 58, 59, 66, 68, 73, and 82 [14,15]. The basic mechanisms of hr-HPV-induced carcinogenesis are that the interactions of the hr-HPV oncoproteins (E6 and E7) with host cellular proteins are involved in the activation or repression of cell cycle progression in carcinogenesis [16]. Previous studies have examined HPV mRNA in oral (oropharyngeal swab or saliva) samples from patients with OPCs and found high rates of HPV E6 oncoproteins and E6/E7 mRNA, which suggests that most patients experience transcriptionally active HPV-related OPCs [17,18]. Survival rates among patients with oral and oropharyngeal cancers appear to be on par with other cancers, with survival for HPV-related OPCs appearing to be better than for non-HPV-related OPCs [19].

Heck’s disease, also known as ‘focal epithelial hyperplasia’, is predominately caused by oral HPV types 13 or 32 and has been reported among Indigenous people worldwide [20,21,22,23,24]. Clinically, Heck’s disease presents as soft, distinct, mostly multiple, smooth papules or nodules having the same colour as the surrounding mucosal epithelium. Although it is a self-limiting and resolving condition, if the condition persists, it can significantly affect the quality of an individual’s life [25].

Worldwide, Indigenous populations experience the highest incidence of OPCs due to societal inequality and disparities in health services [26]. The first peoples of Australia, Aboriginal, and Torres Strait Islanders (hereafter respectfully termed ‘Indigenous’), comprise 3.3% of the total Australian population [27]. The limited evidence suggests that there are higher rates of oral and oropharyngeal cancers among Indigenous people relative to non-Indigenous people in Australian [28], although the HPV attributable fraction remains unknown.

It is necessary to estimate the prevalent and incident infection with disease-associated HPV types to better understand the characteristics and pattern of oral HPV infection among Indigenous Australians. However, to date, most studies examining oral HPV infection have been cross-sectional in design. Our previous studies [29,30] have shown that Indigenous Australian adults had a higher prevalence, incidence, and persistence and a lower clearance of oral HPV infection at baseline and across 12 months than the overall Australian population [31]. This study aims to (1) estimate the prevalence, incidence, persistence, and clearance of any oral HPV type and (2) identify the risk factors associated with oral HPV infection among a cohort of Indigenous Australian adults over a 24-month period.

## 2. Materials and Methods

This study is reported according to STROBE (Strengthening the Reporting of Observational Studies in Epidemiology) guidelines.

### 2.1. Study Design, Sample Selection and Data Collection

The project implemented a longitudinal cohort study design. In February 2018 and January 2019 (baseline), a large convenience sample (n = 1011) of Indigenous Australian adults in South Australia aged 18+ years was recruited through Aboriginal Community Controlled Health Organisations (ACCHOs) who were additional key stakeholders in the study. This study had the oversight of ACCHOs and an Indigenous Reference Group (IRG), who advised on all aspects of project staff employment, participant recruitment, data collection, analysis, and feedback.

The study received ethical approval from the University of Adelaide Human Research Ethics Committee (H-2016–246) and the Aboriginal Health Council of South Australia (04–17-729). All participants provided written informed consent.

Figure 1 shows the details of the data collection flow chart. Data collection included face-to-face interviews and saliva sample collection by experienced research officers who were trained and calibrated in their delivery of the study aims. Data collection was supported and evaluated by a senior Indigenous research officer (JH). These visits were repeated at 12 months (February 2019 and January 2020, n = 743) and 24 months (February 2020 and January 2021, n = 803). The loss to follow-up was tracked, and reasons included death, withdrawal (due to COVID-19 restrictions and others), interstate travel, and incarceration.

### 2.2. Self-Reported Data

The self-reported questionnaire included socio-demographic characteristics, self-rated oral and general health, health related behaviour, HPV vaccination status, and sexual behaviours.

Socio-demographic characteristics were ‘Age’, dichotomised two groups (≥37 vs. < 37 years based on the median of 37 years); ‘Gender’ (Male vs. Female); ‘Geographic location’ (‘Non-metropolitan’ vs. ‘Metropolitan’); ‘Highest educational attainment’ (dichotomised into ‘High school or less’ vs. ‘Trade/TAFE/University’; TAFE stands for ‘Technical and Further Education’ and provides training for vocational occupations); ‘Income’ (defined as ‘Job’ vs. ‘Centrelink-welfare support payments’); Ownership of a government-administered health care card (HCC) (Yes vs. No; an HCC is means-tested and enables access to services such as publicly funded dental care); and Car ownership, from question ‘Do you own a car?’, which had two responses (Yes vs. No).

Self-rated general and oral health were assessed from the questions ‘Would you rate your general or oral health as ‘Excellent/Very good/Good’ or ‘Fair/Poor’?

Health-related behaviours included ‘Tobacco smoking statuses’, defined as ‘current smoker’, ‘ex-smoker’, and ‘never smoked’. ‘Recreational drug use’ was classified as ‘Currently use’, ‘Don’t now but used to’ and ‘Never used’.

Self-reported HPV vaccination status was obtained from the question: ‘Have you ever received a vaccination for HPV?’ and three responses (Yes, No, or Don’t know).

Sexual behaviours stemmed from two questions: (1) Have you ever given oral sex? and (2) Have you ever received oral sex? and were dichotomised as ‘Yes’ (ever given or/and received oral sex) vs. ‘No’ (neither given nor received oral sex).

### 2.3. Saliva Samples Testing and Analysis

Saliva samples were collected using Omnigene Discover kits (OM-501; DNA Genotek Inc., Ottawa, ON, Canada) at baseline and 12 and 24 months, from which microbial DNA for genotyping was extracted for analysis. Antonsson and colleagues have evaluated three different kits (all semi-automated) for DNA extraction, namely Promega’s Maxwell®-16 Viral Total Nucleic Acid Purification Kit, QIAGEN’s QIAamp Mini Elute Virus Spin Kit and QIAamp Blood DNA Mini Kit (QIAcube) (QIAGEN, Venlo, The Netherlands). β-globin polymerase chain reaction (PCR) with the primers PCO3/4 was carried out on all samples to ensure that enough cells to detect human DNA were present and that PCR inhibiting agents were absent. Saliva samples that were β-globin-positive were included in the data analysis. β-globin is a test of DNA integrity; any samples with a negative β-globin were invalid. HPV analysis was performed using a nested PCR system MY09/11 and GP5+/6+ to detect a large spectrum of mucosal HPV type, which included Heck’s-disease-related HPV type and all high-risk HPV types that have oncogenic potential in mucosal tissue. All HPV-DNA-positive samples were sequenced to confirm viral DNA sequences. For the sequencing, HPV-positive PCR products were purified with the Agencourt^®^ AMPure PCR purification kit in a magnetic 96-ring SPRIplate^®^. Sequencing reactions were performed by containing the purified PCR products together with GP+ primer and BigDye Terminator. Direct sequencing was carried out initially. Samples with multiple HPV types were cloned before sequencing, with at least 5 clones sequenced per sample. Sequence reactions were analysed with an automated DNA sequencer (ABI model 3100) (QIAGEN, Venlo, The Netherlands). The DNA sequences were compared with available sequences in GenBank through the BLAST server. The standard PCR method has been used in several projects before by us and others, and it has been proven to be reliable and reproducible [3,29,30].

### 2.4. Statistical Analysis

Data files were managed and summary variables were computed using SAS software version 9.4 (SAS 9.4, SAS Institute Inc., Cary, NC, USA).

Analysis began with statistics describing the point prevalence of oral HPV infection, including any type of HPV infection; HPV 13 or 32; HPV16 or 18; and all high-risk HPV (hr-HPV) infections. hr-HPVs were defined as HPV 16, 18, 26, 31, 33, 35, 39, 45, 51, 52, 53, 56, 58, 59, 66, 68, 73, and 82; they were assessed at three time points (baseline, 12-months, and 24-months).

After that, the incidence, persistence, and clearance of oral HPV infection at baseline and at 12- and 24-months were calculated. For the incidence, persistence, and clearance analyses, we included only participants who not only supplied three saliva samples but also had valid (β-globin positive) samples at all three time points. Incidence was defined as a negative sample at baseline or negative sample at baseline and 12 months, followed by a positive sample at 24 months; persistence was defined as a positive sample at baseline, 12 months, and 24 months (it could be 1, 2, or 3 different HPV types in the same individual); and clearance was defined as a positive sample at baseline or a positive sample at baseline and 12 months, followed by a negative sample at 24 months. Two other conditions were defined as ‘fluctuation’: scenario I was defined as a negative sample at baseline, followed by a positive at 12 months, followed by a negative at 24 months; and scenario II (also named ‘relapsed’ case) was defined as a positive sample at baseline, followed by a negative sample at 12 months, and then, followed by a positive sample at 24 months. Both scenarios I and II were calculated as incidence and clearance of oral HPV infection. Participants lost to follow-up at 12- and/or 24-months or those that were β-globin-negative at one of three time points (named invalid samples) were excluded in this analysis. However, to ensure that the level of missingness or invalid samples did not affect the validity of results, the distribution of baseline characteristics between the two groups was compared.

Period prevalence, which refers to prevalence measured over an interval of time, was estimated using the formula:$$\mathrm{Period}\;\mathrm{Prevalence}=\frac{\mathrm{Persons}\;\mathrm{having}\;a\;\mathrm{particular}\;\mathrm{disease}\;\mathrm{or}\;\mathrm{attribute}\;\mathrm{during}\;a\;\mathrm{given}\;\mathrm{time}\;\mathrm{period}}{\mathrm{Population}\;\mathrm{during}\;\mathrm{the}\;\mathrm{same}\;\mathrm{time}\;\mathrm{period}\;}$$

Statistically significant differences were denoted by 95% confidence intervals (CI) that did not overlap.

Multivariable logistic regression models with binomial distribution estimation were generated to calculate risk indicators. Unadjusted and adjusted prevalence ratios (PRs) and their 95% CI were calculated for prevalence, incidence, persistent, and clearance of any type of HPV infection, HPV 13 or 32, HPV16 or 18, and all high-risk HPV (hr-HPV) infections. Blocks of covariates were entered into multivariable models in five steps: model 1 was the crude model; the sociodemographic factors (age, gender, geographic location, education level, income, health care card holder, car ownership) were entered in model 2; health-related behaviours (smoking and recreational drug status and sexual behaviours) were added into model 3; self-rated general and oral health were added into model 4; and self-reported HPV vaccination status was added into the final model (full model), which was model 5. It is important to note that the final model was built based on a priori selection of covariates as opposed to covariate selection based upon bivariate statistics.

## 3. Results

A total of 1009 participants had at least 1 oral HPV test: 993 of which had a valid test result (at least 1 time point with a β-globin-positive saliva sample); 604 of 993 provided saliva samples at all three time points, and 473 of the 604 had valid (β-globin-positive) samples at all three time points. Of the 993 participants with at least one time point with a β-globin-positive saliva sample, 509 were positive for at least one HPV type at one or more time points.

Table 1 shows baseline characteristics (n = 993) among Indigenous Australian adults. A higher proportion of participants with any oral HPV infection was female (66%), resided in non-metropolitan locations (63%), reported a highest level of education attainment as high school or less (68%), received their income from Centrelink (76%), had an HCC (79%), owned their own car (55%), currently smoked tobacco (nearly 60%), never used recreational drug (46%), had given or received oral sex (approximately 70%), and self-rated excellent/very good/good general health (78%) and oral health (67%). More than 8% had been vaccinated against HPV, and 34% did not know their vaccination status.

The baseline characteristics across three time points of valid and invalid saliva samples are summarised in Appendix A. The valid and invalid samples largely shared the same distribution of baseline characteristics. The exceptions were gender and recreational drug use.

Table 2 presents the HPV type and prevalence of oral HPV infection at three time points. A total of 44 HPV types were identified: 3, 6, 10, 11, 13, 16, 18, 26, 30, 31, 32, 33, 34, 35, 38, 39, 40, 42, 44, 45, 51, 52, 53, 54, 56, 58, 59, 62, 66, 67, 68, 69, 72, 73, 76, 81, 82, 84, 87, 89, 90, 103, 106, 107, 124, 158, 170, and 187. The prevalence of oral HPV infection (any type) was 35.3%, 43.5%, and 24.8% at baseline, 12 months, and 24 months, respectively. The highest prevalent HPV types were those associated with Heck’s disease (HPV 13 or 32) (22.8%, 33.5%, and 15.6%, respectively); the next highest prevalent types were hr-HPV (any hr-HPV type: 16, 18, 26, 31, 33, 35, 39, 45, 51, 52, 53, 56, 58, 59, 66, 68, 73, and 82), at 7.8%, 7.4%, and 4.0%, respectively; and the third highest prevalence types were HPV 16 or 18 (3.3%, 2.5% and 1.6%, respectively) at baseline, 12 months, and 24 months.

The incidence, persistence, and clearance of oral HPV infection at baseline, 12-month, and 24-month follow-up among Indigenous Australian adults are presented in Table 3. For those 473 who had valid HPV tests (β-globin-positive) across all three time points, approximately 45% had no oral HPV infection at any time point (baseline, 12-month, or 24-month follow-up). For those 264 cases who had any HPV-type infection at least once, more than 30% had a new oral HPV infection (including 3.2% relapsed cases) at 12- or 24-month follow-ups; more than 10% had a persistent oral HPV infection at three time points—it could be the same HPV type (45 participants with 12 HPV types, or 2 to 3 different HPV types (11 participants with 15 HPV types) in the same individual) (see Appendix A); and nearly 35% had oral HPV clearance from baseline to 12 or 24 months. For those 185 who had HPV 13 or 32 infections across three time points at least once, approximately 30% had new oral HPV 13 or 32 infections (including more than 9% relapsed cases); about 7% had persistent infections; and more than 25% had oral HPV 13 or 32 clearance from baseline to the 12- or/and 24-month follow-ups. For those 40 who had HPV 16 or 18 infections across three time points at least once, more than 7% had new oral HPV 16 or 18 infections (including more than 6% relapsed cases); 1% had persistent infections; and nearly 7% had oral HPV 16 or 18 clearance from baseline to the 12- or/and 24-month follow-ups. For those 59 who had hr-HPV infections across three time points at least once, more than 9% (including 6% relapsed cases) had new oral high-risk HPV infections; nearly 3% had persistent infections; and approximately 9% had oral hr-HPV clearance.

Also, Table 1 presents the prevalence of HPV infection among Indigenous Australian adults. A higher prevalence of any oral HPV infection was found among those who self-rated excellent/very good/good general health (54%). A higher prevalence of oral HPV 13 or 32 infection was found among those who resided in non-metropolitan locations (approximately 45%), did not own their own car (more than 40%), had not given or received oral sex (nearly 45%), and self-rated excellent/very good/good general health (40%). The prevalence of oral HPV 16 or 18 infection was higher among those residing in metropolitan locations (8%). The prevalence of oral hr-HPV infection was higher among those residing in metropolitan locations (nearly 20%) and those who had given or received oral sex (more than 10%).

The multivariable analyses of any oral HPV infection prevalence are shown in Table 4. A higher prevalence of any oral HPV infection was observed among participants who did not own their own car (PR = 1.16, 95% CI: 1.03–1.31) and self-rated excellent/very good/good general health (PR = 1.25, 95% CI: 1.05–1.48).

The multivariable analyses of oral HPV 13 or 32, 16 or 18, and hr-HPV infection prevalence are shown in Appendix A, respectively. In the HPV 13 or 32 infection model (Appendix A), a higher prevalence was observed among those who resided in non-metropolitan locations (PR = 1.73, 95% CI: 1.42–2.10), did not own their own car (PR = 1.27, 95% CI: 1.08–1.49), and self-rated excellent/very good/good general health (PR = 1.46, 95% CI: 1.15–1.84); and lower prevalence was observed among those who had given or received oral sex (PR = 0.77, 95% CI: 0.64–0.91). After adjusting for all covariates, those who resided in non-metropolitan locations had HPV 13 or 32 infections at a prevalence of nearly two times (PR = 1.81, 95% CI: 1.43–2.28) that of those residing in metropolitan locations. On the contrary, in the HPV 16 or 18 infection model, those who resided in non-metropolitan locations had lower (PR = 0.29, 95% CI: 0.14–0.56) rates of HPV 16 or 18 infections than their metropolitan-dwelling counterparts (Appendix A). In the hr-HPV infection model (Appendix A), lower prevalence of oral hr-HPV infection was observed among those who resided in non-metropolitan locations (PR = 0.28, 95% CI: 0.20–0.42), and higher prevalence of oral hr-HPV infections was found in ex-users of recreational drugs (PR = 1.77, 95% CI: 1.18–2.67) and those having given or received oral sex (PR = 1.58, 95% CI: 1.48–2.28). After adjusting for all covariates, those who resided in non-metropolitan locations had lower (PR = 0.29, 95% CI: 0.19–0.44) rates of infection than their metropolitan-dwelling counterparts, but participants who were currently users or ex-users of recreational drugs had hr-HPV infection rates that were nearly two times higher (PR = 2.01, 95% CI: 1.23–3.28 and PR = 1.75, 95% CI: 1.22–3.04, respectively) than those who had never used drugs. Participants who had given or received oral sex had nearly 1.2 times higher prevalence (PR = 1.16, 95% CI: 1.06–1.63) than those with non-hr-HPV infections.

The sociodemographic characteristics, general and oral health, and health-related behaviours of participants (n = 473) who had not or had incidence, persistent, or clearance of any oral HPV infection from baseline to the 24 months follow-up are presented in Table 5. A higher incidence of any oral HPV infection was found among those with high school or less as the highest level of education attainment (nearly 35%), those that did not own their own car (more than 35%), and ex-smokers (more than 40%); a higher persistence of oral HPV infection was observed among those who resided in non-metropolitan locations (nearly 15%); and a higher clearance of any oral HPV infection was found in those who did not have their own car (more than 35%) and self-rated excellent/very good/good general health (more than 35%).

The multivariable analyses of incidence, persistence, or clearance of any oral HPV infection from baseline to the 24 months follow-up are presented in Table 6 and Appendix A. In the incidence model (Table 6), a higher incidence of any oral HPV infection was observed among participants who had high school or less as the highest level of education attainment (PR = 1.33, 95% CI: 1.01–1.80), did not own their own car (PR = 1.29, 95% CI: 1.00–1.69), and were ex-smokers (PR = 1.64, 95% CI: 1.07–2.51). After adjusting for all covariates, participants who were ex-smokers were 1.5 times (PR = 1.53, 95% CI: 1.02–2.53) more susceptible to getting a new or relapsed infection at 12 or 24 months. In the persistent model (Appendix A), participants residing in non-metropolitan locations were 1.5 times (PR = 1.53, 95% CI: 1.00–2.63) more likely to show persistent oral HPV infections. In the clearance model (Appendix A), participants who did not own their own car and reported excellent/very good/good general health had 1.2 times higher clearance of oral HPV infections (PR = 1.26, 95% CI: 1.01–1.62 and PR = 1.28, 95% CI: 1.02–1.79), respectively, than their counterparts. After adjusting for social demographic characteristics (model 2), participants who did not own their own car had more than 1.3 times (PR = 1.34, 95% CI: 1.00–1.79) higher clearance of oral HPV infection than car owners at 12 or 24 months.

## 4. Discussion

To the best of our knowledge, this is the first study to provide the natural history of oral HPV infection among Indigenous South Australians across 24 months; it is also the first study to track the incidence and persistence of HPV infections associated with Heck’s disease. This was achieved through the IRG and ACCHOs being actively involved in and supportive of this study, including recruitment and training of research officers.

Our findings indicate that current and previous recreational drug use and oral sexual behaviours were positively associated with high-risk HPV infections. Indigenous South Australian adults who reside in non-metropolitan locations had a higher risk of acquiring HPV 13 or 32 infections, but a lower risk of HPV 16 or 18 and hr-HPV infections. Other risk indicators associated with HPV 13 or 32 infections included non-ownership of a car and fair/poor self-rated general health. Meanwhile, we found that the risk indicators associated with incidence of oral HPV infection were lower education levels and non-ownership of a car; risk indicators associated with persistent oral HPV infection were residing in non-metropolitan locations; and a risk indicator associated with clearance of oral HPV infection was also non-ownership of a car.

Compared with our previous two studies [29,30], a new HPV type, hr-HPV 26, has been added, and 15 existed HPV types have been cleared, including 5 hr-HPV 31, 33, 39, 59, and 68 at a follow-up time point of 24 months. Therefore, the prevalence of oral HPV has decreased from more than 40% to approximately 25%; in particular, the prevalence of oral hr-HPV infection has decreased by 50% (from 8% to 4%).

Evidence has shown that the HPV vaccine is effective in reducing the prevalence of oral HPV infection [31]. In Australia, however, school-based HPV vaccination programs reported that completed HPV vaccine coverage by age 15 was lower amongst Indigenous adolescents (71.5%) compared with overall Australian adolescents (79.1%) in 2020 [32,33]. Given the high prevalence and incidence of oral HPV infection and low proportion of HPV-vaccinated participants in our study, there is an urgent need to ensure high HPV vaccination coverage among all Indigenous Australians [31]. To close the gap or reduce the inequality, it is important to work with Indigenous communities and organisations to develop and lead culturally appropriate solutions, thereby, maintaining and increasing participation in vaccination for Indigenous Australians [33].

Our findings demonstrated that acquisition of oral oncogenic HPV was significantly associated with oral sex. This finding keeps in line with previous studies [34,35,36]. Also, D’Souza and colleagues’ study shows that performing oral sex on a woman may confer a higher risk of oral HPV acquisition than performing oral sex on a man [37]. At the same time, we observed that the prevalence of hr-HPV and HPV 16/18 decreased from 0 to 24 months, which could be due to changes in sexual behaviours during the COVID-19 pandemic. In addition, our finding was consistent with a previous study [38] which has shown that recreational drug consumption is more common among those with sexually transmitted infections (STIs) including HPV infection. There was no association between oral HPV infection at 24 months and age, gender, or smoking consumption.

The incidence of HPV types associated with Heck’s disease was much higher among Indigenous Australian adults (16.1%) than non-Indigenous Australian young adults (3.2% HPV 32 and 0% HPV 13) [39], which may be associated with Indigenous genetic susceptibility, such as the human lymphocytic antigen (HLA)-DR4(DRB1*0404) allele occurring frequently in Indigenous populations of the Americas [40]. However, there has been no documentation of Heck’s disease among the Indigenous Australians included in this cohort, especially followed up over three time points (the COVID-19 pandemic meant that oral examinations were not possible).

Compared with a study of non-Indigenous Australians [39], our study found a higher incidence rate (25% vs. 30%) and lower clearance rates (79% vs. 33%) of oral HPV infection. Meanwhile, compared with previous studies [41,42], Indigenous Australian adults had a higher persistence oral hr-HPV infection, especially oral HPV 16 or/and 18 infections with more than 10% persistence and <9% clearance. A total of 40 out of 59 (=68%) cases of hr-HPV was due to HPV 16/18 infection. hr-HPV associated with OPCs, given such high persistence and low clearance, indicated that Indigenous Australians have higher risks of hr-HPV related OPCs.

Strengths of this study include: (1) engagement with the Indigenous Australian community through partnerships and involvement of the study’s Indigenous Reference Group; (2) a large representation of the broader South Australian Indigenous population (n = 1011), with the findings potentially having meaning for other Indigenous groups residing in similar socio-economic regions of the world; and (3) a high response rate: 73.5% and 79.4% in the 12- and 24-month follow-ups, respectively. Limitations include (1) lack of oral clinical examination data due to the COVID-19 pandemic: this may not only influence the outcomes of the study, but also meant that we did not have the ability to correlate clinical oral disease statuses, such as Heck’s disease and early stage OPCs; (2) non-representative findings, with a higher proportion of females, those residing in non-metropolitan locations, welfare-based income, health care card holders, and current smokers than in broader Indigenous groups [43]; (3) loss to follow-up across the 12- and 24-month time points, which might lead to biased estimates; and (4) combined sex behaviours (given and received oral sex) may not be accurate as another study demonstrated the highest risk from giving cunnilingus [37].

This study could inform guidelines for the prevention and control of oral HPV infections and help to identify the factors associated with both infection and clearance; it may explore a new potential test for early detection of individuals at risk of developing oropharyngeal cancer. Therefore, oral HPV DNA tests could be considered as a potential screening test for oropharyngeal cancer prevention in the future.

## 5. Conclusions

Our findings indicated that the prevalence and incidence of oral HPV infection in a large sample of Indigenous Australians were high. Engaging in oral sexual behaviours and recreational drug use are risk factors associated with high-risk HPV infections. Not owning a car and fair/poor self-rated general health are risk indictors associated with oral HPV 13 or 32 infections. These findings support current efforts to increase HPV vaccination uptake amongst Indigenous Australian adolescents.

## Figures and Tables

**Figure 1 viruses-15-01573-f001:**
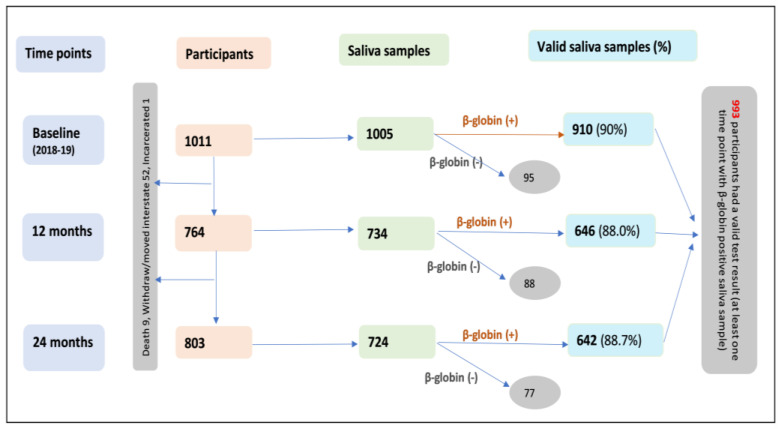
Flow chart of data collection at baseline (2018–2019), 12, and 24 months among Indigenous Australian adults.

**Table 1 viruses-15-01573-t001:** Baseline sample characteristics and prevalence of oral HPV infection among Indigenous Australian adults (n = 993).

	Overall		Any HPV Infection		HPV 13/32 Infection		HPV 16/18 Infection		hr-HPV Infection	
	n	% (95% CI)	n	% (95% CI)	n	% (95% CI)	n	% (95% CI)	n	% (95% CI)
Total	993		509	51.3 (48.1–54.4)	371	37.4 (34.3–40.4)	45	4.5 (3.2–5.8)	107	10.8 (8.9–12.7)
Age group (years)										
≥37	518	52.2 (49.2–55.4)	271	52.3 (48.0–56.6)	191	36.9 (32.7–41.0)	24	4.6 (2.8–6.4)	55	10.6 (8.0–13.3)
<37	475	47.8 (44.6–50.8)	238	50.1 (45.6–54.6)	180	37.9 (33.5–42.3)	21	4.4 (2.6–6.3)	52	10.9 (8.1–13.8)
Missing	0		0		0		0		0	
Sex										
Male	334	**33.6 (30.8–36.6)**	171	51.2 (45.8–56.6)	119	35.6 (30.5–40.8)	15	4.5 (2.3–6.7)	39	11.7 (8.3–15.2)
Female	659	**66.4 (63.4–69.2)**	338	51.3 (47.5–55.1)	252	38.2 (34.5–42.0)	30	4.6 (3.0–6.1)	68	10.3 (8.0–12.6)
Missing	0		0		0		0		0	
Geographic location										
Non-metropolitan	622	**62.8 (59.8–65.8)**	333	53.5 (49.6–57.5)	276	**44.4 (40.5–48.3)**	16	**2.6 (1.3–3.8)**	35	**5.6 (3.8–7.5)**
Metropolitan	369	**37.2 (34.2–40.2)**	176	47.7 (42.6–52.8)	95	**25.7 (21.3–30.2)**	29	**7.9 (5.1–10.6)**	72	**19.5 (15.5–23.6)**
Missing	2		0		0		0		0	
Education level										
High school or less	667	**68.2 (65.3–71.1)**	348	52.2 (48.4–56.0)	258	38.7 (35.0–42.4)	26	3.9 (2.4–5.4)	67	10.1 (7.8–12.3)
Trade or over	311	**31.8 (28.9–34.7)**	157	50.5 (44.9–56.0)	111	35.7 (30.4–41.0)	19	6.1 (3.4–8.8)	39	12.5 (8.9–16.2)
Missing	15		4		2		0		1	
Income										
Centrelink	743	**75.9 (73.4–78.7)**	393	52.9 (49.3–56.5)	281	37.8 (34.3–41.3)	35	4.7 (3.2–6.2)	82	11.1 (8.8–13.3)
Job	236	**24.1 (21.3–26.6)**	111	47.0 (40.7–53.4)	86	36.4 (30.3–42.6)	10	4.2 (1.7–6.8)	25	10.6 (6.7–14.5)
Missing	14		5		4		0		0	
Health care card										
Yes	745	**78.7 (76.4–81.6)**	386	51.8 (48.2–55.4)	276	37.0 (33.6–40.5)	35	4.7 (3.2–6.2)	86	11.6 (9.3–13.9)
No	202	**21.3 (18.4–23.6)**	103	51.0 (44.1–57.9)	78	38.6 (31.9–45.3)	8	4.0 (1.3–6.7)	19	9.4 (5.4–13.4)
Missing	46		20		17		2		2	
Car ownership										
No	440	**44.7 (41.4–47.6)**	245	55.7 (51.0–60.3)	187	**42.5 (37.9–47.1)**	17	3.9 (2.1–5.7)	39	8.9 (6.2–11.5)
Yes	544	**55.3 (52.4–58.6)**	261	48.0 (43.8–52.2)	182	**33.5 (29.5–37.4)**	28	5.1 (3.3–7.0)	68	12.5 (9.7–15.3)
Missing	9		3		2		0		0	
Smoke status										
Current smoker	556	**59.2 (56.2–62.5)**	270	48.6 (44.4–52.7)	194	34.7 (30.7–38.7)	19	3.4 (1.9–4.9)	53	9.5 (7.1–12.0)
Ex-smoker	113	**12.0 (9.8–13.9)**	67	59.3 (50.2–68.4)	46	40.7 (31.6–49.8)	7	6.2 (1.7–10.6)	19	16.8 (9.9–23.7)
Never smoker	270	**28.8 (26.0–31.7)**	142	52.6 (46.6–58.6)	108	40.0 (34.1–45.9)	13	4.8 (2.3–7.4)	28	10.4 (6.8–14.1)
Missing	54		30		23		6		7	
Recreational drug use										
Current user	205	**20.9 (18.3–23.4)**	111	54.1 (47.3–61.0)	79	38.5 (31.9–45.2)	6	2.9 (0.6–5.2)	24	11.7 (7.3–16.1)
Ex-user	329	**33.6 (30.5–36.4)**	166	50.5 (45.0–55.9)	108	32.8 (27.7–37.9)	20	6.1 (3.5–8.7)	47	14.3 (10.5–18.1)
Never use	445	**45.5 (42.6–48.8)**	224	50.3 (45.7–55.0)	177	39.8 (35.2–44.3)	19	4.3 (2.4–6.2)	36	8.1 (5.6–10.6)
Missing	14		8		7		0		0	
Had oral sex										
Yes	620	**69.6 (66.2–72.2)**	304	49.0 (45.1–53.0)	208	**33.5 (29.8–37.2)**	28	4.5 (2.9–6.2)	76	**12.3 (10.5.–14.8)**
No	271	**30.4 (27.8–33.8)**	146	53.9 (47.9–59.8)	118	**43.5 (37.6–49.5)**	12	4.4 (2.0–6.9)	21	**7.8 (4.6–10.0)**
Missing	102		59		45		5		10	
Self-rated general health										
Fair/Poor	214	**22.0 (19.7–24.8)**	408	**53.7 (50.1–57.2)**	305	**40.1 (36.6–43.6)**	36	4.7 (3.2–6.2)	79	10.4 (8.2–12.6)
Excel/very good/Good	760	**78.0 (75.2–80.3)**	92	**43.0 (36.3–49.6)**	59	**27.6 (21.6–33.6)**	9	4.2 (1.5–6.9)	27	12.6 (8.2–17.1)
Missing	19		9		7		0		1	
Self-rated oral health										
Fair/Poor	642	**33.5 (30.5–36.4)**	342	53.3 (49.4–57.1)	246	38.3 (34.6–42.1)	31	4.8 (3.2–6.5)	71	7.9 (5.8–10.0)
Excel/very good/Good	323	**66.5 (63.6–69.5)**	155	48.0 (42.5–53.4)	115	35.6 (30.4–40.8)	14	4.3 (2.1–6.6)	35	8.7 (5.6–11.7)
Missing	28		12		0		0		1	
Ever received HPV vaccination										
Yes	82	**8.3 (6.6–10.1)**	39	47.6 (36.7–58.4)	28	34.1 (23.9–44.4)	5	6.1 (0.9–11.3)	10	12.2 (5.1–19.3)
No	565	**57.4 (54.3–60.5)**	302	53.5 (49.3–57.6)	227	40.2 (36.1–44.2)	23	4.1 (2.4–5.7)	55	9.7 (7.3–12.2)
Do not know	337	**34.2 (31.3–37.2)**	162	48.2 (42.9–53.6)	113	33.6 (28.6–38.7)	17	5.1 (2.7–7.4)	41	12.2 (8.7–15.7)
Missing	9		6		3		0		1	

Note: Bold were donated as statistically significant.

**Table 2 viruses-15-01573-t002:** Oral HPV type among Indigenous Australian adults.

	Baseline		12 Months		24 Months	
	n	% (95% CI)	n	% (95% CI)	n	% (95% CI)
Total	910	100 (100.0–100.0)	645	100 (100.0–100.0)	642	100 (100.0–100.0)
HPV positive ≥ 1	321	35.3 (32.2–38.4)	280	43.4 (39.6–47.2)	159	24.8 (21.4–28.1)
hr-HPV *	71	7.8 (6.1–9.5)	48	7.4 (5.4–9.5)	26	4.0 (2.5–5.6)
HPV type						
3	2	0.2 (0.0–0.5)	0	0.0 (0.0–0.0)	1	0.2 (0.0–0.5)
6	3	0.3 (0.0–0.7)	2	0.3 (0.0–0.7)	1	0.2 (0.0–0.5)
7	1	0.1 (0.0–0.3)	0	0.0 (0.0–0.0)	0	0.0 (0.0–0.0)
10	1	0.1 (0.0–0.3)	2	0.3 (0.0–0.7)	0	0.0 (0.0–0.0)
11	0	0.0 (0.0–0.0)	1	0.2 (0.0–0.5)	1	0.2 (0.0–0.5)
13	88	9.7 (7.7–11.6)	61	9.5 (7.2–11.7)	39	6.1 (4.2–7.9)
16	14	1.5 (0.7–2.3)	14	2.2 (1.0–3.3)	7	1.1 (0.3–1.9)
18	16	1.8 (0.9–2.6)	2	0.3 (0.0–0.7)	3	0.5 (0.0–1.0)
26	0	0.0 (0.0–0.0)	0	0.0 (0.0–0.0)	1	0.2 (0.0–0.5)
30	2	0.2 (0.0–0.5)	0	0.0 (0.0–0.0)	0	0.0 (0.0–0.0)
31	1	0.1 (0.0–0.3)	0	0.0 (0.0–0.0)	0	0.0 (0.0–0.0)
32	119	13.1 (10.9–15.3)	155	24.0 (20.7–27.3)	61	9.5 (7.2–11.8)
33	1	0.1 (0.0–0.3)	1	0.2 (0.0–0.5)	0	0.0 (0.0–0.0)
34	1	0.1 (0.0–0.3)	1	0.2 (0.0–0.5)	0	0.0 (0.0–0.0)
35	3	0.3 (0.0–0.7)	2	0.3 (0.0–0.7)	2	0.3 (0.0–0.7)
39	2	0.2 (0.0–0.5)	1	0.2 (0.0–0.5)	0	0.0 (0.0–0.0)
40	1	0.1 (0.0–0.3)	0	0.0 (0.0–0.0)	0	0.0 (0.0–0.0)
42	1	0.1 (0.0–0.3)	0	0.0 (0.0–0.0)	10	1.6 (0.6–2.5)
44	1	0.1 (0.0–0.3)	0	0.0 (0.0–0.0)	0	0.0 (0.0–0.0)
45	3	0.3 (0.0–0.7)	4	0.6 (0.0–1.2)	1	0.2 (0.0–0.5)
51	1	0.1 (0.0–0.3)	2	0.3 (0.0–0.7)	1	0.2 (0.0–0.5)
52	2	0.2 (0.0–0.5)	0	0.0 (0.0–0.0)	1	0.2 (0.0–0.5)
53	2	0.2 (0.0–0.5)	3	0.5 (0.0–1.0)	1	0.2 (0.0–0.5)
54	1	0.1 (0.0–0.3)	1	0.2 (0.0–0.5)	1	0.2 (0.0–0.5)
56	4	0.4 (0.0–0.9)	7	1.1 (0.3–1.9)	1	0.2 (0.0–0.5)
58	5	0.5 (0.1–1.0)	0	0.0 (0.0–0.0)	2	0.3 (0.0–0.7)
59	5	0.5 (0.1–1.0)	1	0.2 (0.0–0.5)	0	0.0 (0.0–0.0)
62	1	0.1 (0.0–0.3)	2	0.3 (0.0–0.7)	5	0.8 (0.1–1.4)
66	9	1.0 (0.3–1.6)	10	1.6 (0.3–1.9)	4	0.6 (0.0–1.2)
67	2	0.2 (0.0–0.5)	0	0.0 (0.0–0.0)	0	0.0 (0.0–0.0)
68	1	0.1 (0.0–0.3)	0	0.0 (0.0–0.0)	0	0.0 (0.0–0.0)
69	8	0.9 (0.3–1.5)	0	0.0 (0.0–0.0)	2	0.3 (0.0–0.7)
72	7	0.8 (0.2–1.3)	7	1.1 (0.3–1.9)	3	0.5 (0.0–1.0)
73	1	0.1 (0.0–0.3)	1	0.2 (0.0–0.5)	1	0.2 (0.0–0.5)
76	0	0.0 (0.0–0.0)	0	0.0 (0.0–0.0)	1	0.2 (0.0–0.5)
81	3	0.3 (0.0–0.7)	0	0.0 (0.0–0.0)	1	0.2 (0.0–0.5)
82	1	0.1 (0.0–0.3)	0	0.0 (0.0–0.0)	1	0.2 (0.0–0.5)
84	1	0.1 (0.0–0.3)	0	0.0 (0.0–0.0)	1	0.2 (0.0–0.5)
87	1	0.1 (0.0–0.3)	0	0.0 (0.0–0.0)	0	0.0 (0.0–0.0)
89	0	0.0 (0.0–0.0)	0	0.0 (0.0–0.0)	1	0.2 (0.0–0.5)
90	5	0.5 (0.1–1.0)	0	0.0 (0.0–0.0)	0	0.0 (0.0–0.0)
106	1	0.1 (0.0–0.3)	0	0.0 (0.0–0.0)	0	0.0 (0.0–0.0)
107	0	0.0 (0.0–0.0)	0	0.0 (0.0–0.0)	1	0.2 (0.0–0.5)
182	0	0.0 (0.0–0.0)	0	0.0 (0.0–0.0)	2	0.3 (0.0–0.7)

Note: * hr-HPV type = 16, 18, 26, 31, 33, 35, 39, 45, 51, 52, 53, 56, 58, 59, 66, 68, 73 and 82.

**Table 3 viruses-15-01573-t003:** Incidence, persistence, and clearance of oral HPV infection at baseline, 12-month follow-up, and 24-month follow-up among Indigenous Australian adults (n = 473).

	No Oral HPV Infection (n = 209)			n	% (95% CI)		Total
	Baseline	12 Months	24 Months			n	% (95% CI)
No infection	×	×	×			209	44.2 (39.7–48.7)
	Any oral HPV infection (n = 264)			n	% (95% CI)	n	% (95% CI)
	Baseline	12 months	24 months				
Incidence	×	√ or ×	√	51	10.8 (8.0–13.6)	145	30.7 (26.5–34.8)
Fluctuation	×	√	×	79	16.7 (13.6–20.3)		
	√	x	√	15	3.2 (1.9–5.2)		
Clearance	√	× or √	×	63	13.3 (10.2–16.4)	157	33.2 (28.9–37.5)
Persistence	√	√	√			56	11.8 (8.9–14.8)
	Oral HPV 13 or 32 infection (n = 185)			n	% (95% CI)	n	% (95% CI)
	Baseline	12 months	24 months				
Incidence	×	√ or ×	√	30	6.3 (4.5–8.9)	140	29.6 (25.7–33.9)
Fluctuation	×	√	×	67	14.2 (11.3–17.6)		
	√	×	√	43	9.1 (6.8–12.0)		
Clearance	√	× or √	×	14	3.0 (1.8–4.9)	124	26.2 (22.5–30.4)
Persistence	√	√	√			31	6.6 (4.7–9.2)
	Oral HPV 16 or 18 infection (n = 40)			n	% (95% CI)	n	% (95% CI)
	Baseline	12 months	24 months				
Incidence	×	√ or ×	√	2	0.4 (0.1–1.5)	35	7.4 (5.4–10.1)
Fluctuation	×	√	×	4	0.9 (0.3–2.2))		
	√	x	√	29	6.1 (4.3–8.7)		
Clearance	√	× or √	×	0	0.0 (0.0–0.0)	33	7.0 (5.0–9.6)
Persistence	√	√	√			5	1.1 (0.5–2.5)
	Any oral hr-HPV infection (n = 59)			n	% (95% CI)	n	% (95% CI)
	Baseline	12 months	24 months				
Incidence	×	√ or ×	√	5	1.1 (0.5–2.5)	44	9.3 (7.0–12.3)
Fluctuation	×	√	×	10	2.1 (1.2–3.9)		
	√	x	√	29	6.1 (4.3–8.7)		
Clearance	√	× or √	×	2	0.4 (0.1–1.5)	41	8.7 (6.5–11.6)
Persistence	√	√	√			13	2.8 (1.6–4.7)

Notes: ‘×’: HPV test negative; ‘√’ HPV test positive; hr-HPV type = 16, 18, 26, 31, 33, 35, 39, 45, 51, 52, 53, 56, 58, 59, 66, 68, 73, and 82. ‘Fluctuation’ was highlighted using the gray colors.

**Table 4 viruses-15-01573-t004:** Multivariable association between oral HPV infection prevalence and risk factors among Indigenous Australian adults (n = 993).

	Model 1	Model 2	Model 3	Model 4	Model 5
	Prevalence Ratio (95% CI)				
Age group (years)					
≥37	1.04 (0.92–1.18)	1.06 (0.93–1.20)	1.02 (0.89–1.18)	1.09 (0.94–1.26)	1.06 (0.91–1.24)
<37	ref	ref	ref	ref	ref
Sex					
Male	1.00 (0.88–1.13)	0.99 (0.87–1.13)	0.96 (0.82–1.12)	0.94 (0.81–1.11)	0.94 (0.80–1.10)
Female	ref	ref	ref	ref	ref
Geographic location					
Non-metropolitan	1.12 (0.98–1.28)	1.11 (0.97–1.27)	1.11 (0.95–1.28)	1.09 (0.93–1.27)	1.11 (0.95–1.30)
Metropolitan	ref	ref	ref	ref	ref
Education level					
High school or less	1.03 (0.90–1.18)	0.99 (0.86–1.14)	0.93 (0.80–1.09)	0.94 (0.81–1.10)	0.93 (0.80–1.09)
Trade or over	ref	ref	ref	ref	ref
Income					
Centrelink	1.12 (0.97–1.31)	1.11 (0.91–1.37)	1.11 (0.88–1.39)	1.10 (0.87–1.38)	1.11 (0.88–1.41)
Job	ref	ref	ref	ref	ref
Health care card					
Yes	1.01 (0.87–1.18)	0.93 (0.77–1.12)	0.92 (0.75–1.12)	0.95 (0.77–1.17)	0.93 (0.76–1.15)
No	ref	ref	ref	ref	ref
Car ownership					
No	**1.16 (1.03–1.31)**	1.11 (0.97–1.28)	1.09 (0.93–1.27)	1.10 (0.94–1.29)	1.10 (0.94–1.29)
Yes	ref	ref	ref	ref	ref
Smoke status					
Current smoker	0.93 (0.80–1.07)		0.87 (0.74–1.03)	0.87 (0.74–1.04)	0.88 (0.74–1.04)
Ex-smoker	1.13 (0.93–1.37)		1.07 (0.86–1.33)	1.05 (0.85–1.31)	1.06 (0.88–1.25)
Never smoker	ref		ref	ref	ref
Recreational drug use					
Current user	1.08 (0.92–1.26)		1.12 (0.92–1.36)	1.14 (0.94–1.38)	1.14 (0.94–1.39)
Ex-user	1.00 (0.87–1.15)		1.05 (0.88–1.24)	1.06 (0.89–1.26)	1.06 (0.89–1.27)
Never use	ref		ref	ref	ref
Had oral sex					
Yes	0.91 (0.80–1.05)		0.93 (0.79–1.07)	0.94 (0.80–1.10)	0.94 (0.80–1.10)
No	ref		ref	ref	ref
Self-rated general health					
Excel/very good/Good	**1.25 (1.05–1.48)**			1.18 (0.96–1.46)	1.17 (0.95–1.45)
Fair/Poor	ref			ref	ref
Self-rated oral health					
Excel/very good/Good	1.11 (0.97–1.27)			1.103 (0.87–1.21)	1.03 (0.88–1.22)
Fair/Poor	ref			ref	ref
Ever received HPV vaccination					
No	1.12 (0.88–1.43)				1.19 (0.91–1.57)
Do not know	1.01 (0.78–1.31)				1.19 (0.89–1.58)
Yes	ref				ref

Notes: Model 1: crude model; Model 2: adjusted for the sociodemographic factors (age, gender, geographic location, education level, income, health care card holder, car ownership); Model 3: plus adjusting for health-related behaviours (smokes and recreational drug status, and sexual behaviours); Model 4: plus adjusting for self-rated general and oral health; Model 5 (full model): plus adjusting for self-reported HPV vaccination status. Bolds are donated as statistically significant differences. Bold were donated as statistically significant.

**Table 5 viruses-15-01573-t005:** Baseline sample characteristics and association with incidence, persistence, and clearance of any oral HPV infection among Indigenous Australian adults (n = 473).

	Overall		Incidence		Persistent		Clearance	
	n	% (95% CI)	n	% (95% CI)	n	% (95% CI)	n	% (95% CI)
Total	473		145	30.7 (26.5–34.9)	56	11.9 (8.9–14.8)	157	33.3 (29.0–37.5)
Age group (years)								
≥37	269	**56.9 (52.4–61.4)**	81	30.2 (24.7–35.7)	37	13.8 (9.7–18.0)	86	32.1 (26.5–37.7)
<37	204	**43.1 (38.6–47.6)**	64	31.4 (25.0–37.8)	19	9.3 (5.3–13.3)	71	34.8 (28.2–41.4)
Missing	0		0		0		0	
Sex								
Male	125	**26.4 (22.4–30.4)**	40	32.3 (24.0–40.5)	12	9.7 (4.5–14.9)	43	34.7 (26.3–43.1)
Female	348	**73.6 (69.6–77.6)**	105	30.2 (25.3–35.0)	44	12.6 (9.1–16.1)	114	32.8 (27.8–37.7)
Missing	0		0		0		0	
Geographic location								
Non-metropolitan	284	**60.0 (55.6–64.4)**	90	31.8 (26.4–37.2)	39	**13.8 (11.2–17.8)**	89	31.4 (26.0–36.9)
Metropolitan	189	**40.0 (35.5–44.4)**	55	29.1 (22.6–35.6)	17	**9.0 (4.9–11.1)**	68	36.0 (29.1–42.8)
Missing	0		0		0		0	
Education level								
High school or less	297	**63.9 (59.5–68.3)**	101	**34.1 (30.3–39.5)**	37	12.5 (8.7–16.3)	95	32.1 (26.8–37.4)
Trade or over	168	**36.1 (31.7–40.5)**	43	**25.6 (19.0–30.2)**	19	11.3 (6.5–16.1)	62	36.9 (29.6–44.2)
Missing	8		1		0		0	
Income								
Centrelink	336	**72.7 (68.7–76.8)**	103	30.7 (25.8–35.7)	43	12.8 (9.2–16.4)	109	32.5 (27.5–37.6)
Job	126	**27.3 (23.2–31.3)**	39	31.0 (22.9–39.1)	13	10.3 (5.0–15.6)	45	35.7 (27.3–44.1)
Missing	11		3		0		3	
Health care card								
Yes	343	**76.6 (72.6–80.5)**	108	31.6 (26.6–36.5)	43	12.6 (9.0–16.1)	109	31.9 (26.9–36.8)
No	105	**23.4 (19.5–27.4)**	30	28.6 (19.9–37.2)	13	12.4 (6.1–18.7)	40	38.1 (28.8–47.4)
Missing	25		7		0		8	
Car ownership								
No	188	**40.2 (35.7–44.6)**	67	**35.6 (33.0–42.5)**	20	10.6 (6.2–15.1)	72	**38.3 (34.2–45.3)**
Yes	280	**59.8 (55.4–64.3)**	77	**27.6 (22.3–32.9)**	36	12.9 (9.0–16.9)	85	**30.5 (25.0–33.9)**
Missing	5		1		0		0	
Smoke status								
Current smoker	259	**57.7 (53.1–62.3)**	77	29.7 (24.1–35.3)	33	12.7 (8.7–16.8)	75	29.0 (23.4–34.5)
Ex-smoker	64	**14.3 (11.0–17.5)**	26	**40.6 (32.5–52.7)**	4	6.3 (0.3–12.2)	26	40.6 (28.5–52.7)
Never smoker	126	**28.1 (23.9–32.2)**	31	**24.8 (17.2–31.4)**	18	14.4 (8.2–20.6)	44	35.2 (26.8–43.6)
Missing	24		11		0		12	
Recreational drug use								
Current user	80	**17.2 (13.7–20.6)**	22	27.5 (17.7–37.3)	12	15.0 (7.1–22.9)	28	35.0 (24.5–45.5)
Ex-user	144	**30.9 (26.7–35.1)**	50	35.0 (27.1–42.8)	12	8.4 (3.8–13.0)	45	31.5 (23.8–39.1)
Never use	242	**51.9 (47.4–56.5)**	71	29.3 (23.6–35.1)	32	13.2 (8.9–17.5)	82	33.9 (27.9–39.9)
Missing	7		2		0		2	
Had oral sex								
Yes	317	**75.3 (71.2–79.4)**	94	29.7 (24.6–34.7)	34	10.7 (7.3–14.1)	105	33.1 (27.9–38.3)
No	104	**24.7 (20.6–28.8)**	36	35.0 (25.7–44.2)	10	9.7 (4.0–15.5)	36	35.0 (25.7–44.2)
Missing	52		15		12		16	
Self-rated general health								
Fair/Poor	112	**24.2 (20.3–28.2)**	28	25.0 (17.0–33.0)	14	12.5 (6.4–18.6)	31	**27.7 (19.4–30.0)**
Excel/very good/Good	350	**75.8 (71.8–79.7)**	114	32.7 (27.7–37.6)	41	11.7 (8.4–15.1)	124	**35.5 (30.5–40.6)**
Missing	11		3		1		2	
Self-rated oral health								
Fair/Poor	175	**38.3 (33.8–42.8)**	56	32.0 (25.1–38.9)	21	12.0 (7.2–16.8)	54	30.9 (24.0–37.7)
Excel/very good/Good	282	**61.7 (57.2–66.2)**	85	30.2 (24.9–35.6)	33	11.7 (8.0–15.5)	98	34.9 (29.3–40.5)
Missing	16		4		2		5	
Ever received HPV vaccination								
Yes	39	**8.4 (5.8–10.9)**	14	35.9 (20.8–51.0)	4	10.3 (0.7–19.8)	12	30.8 (16.2–45.3)
No	260	55.7 (51.2–60.2)	86	33.1 (27.3–38.8)	36	13.8 (9.6–18.1)	90	34.6 (28.8–40.4)
Do not know	168	36.0 (31.6–40.3)	43	25.7 (19.1–32.4)	15	9.0 (4.6–13.3)	54	32.3 (25.2–39.5)
Missing	6		2		0		1	

Notes: Bolds were donated as statistically significant differences.

**Table 6 viruses-15-01573-t006:** Multivariable association between incidence of any oral HPV infection and risk factors among Indigenous Australian adults (n = 473).

	Model 1	Model 2	Model 3	Model 4	Model 5
	Prevalence Ratio (95% CI)				
Age group (years)					
≥37	0.96 (0.73–1.27)	1.06 (0.80–1.41)	1.01 (0.73–1.42)	1.02 (0.72–1.45)	1.01 (0.71–1.44)
<37	ref	ref	ref	ref	ref
Sex					
Male	1.07 (0.79–1.44)	1.04 (0.81–1.45)	1.00 (0.70–1.44)	1.05 (0.73–1.52)	1.08 (0.74–1.56)
Female	ref	ref	ref	ref	ref
Geographic location					
Non-metropolitan	1.09 (0.82–1.45)	1.08 (0.81–1.45)	1.13 (0.82–1.57)	1.13 (0.81–1.58)	1.13 (0.81–1.57)
Metropolitan	ref	ref	ref	ref	ref
Education level					
High school or less	**1.33 (1.01–1.80)**	1.32 (0.95–1.83)	1.29 (0.91–1.84)	1.31 (0.92–1.86)	1.31 (0.92–1.86)
Trade or over	ref	ref	ref	ref	ref
Income					
Centrelink	0.99 (0.73–1.35)	0.81 (0.53–1.23)	0.68 (0.42–1.10)	0.67 (0.40–1.11)	0.68 (0.41–1.12)
Job	ref	ref	ref	ref	ref
Health care card					
Yes	1.11 (0.79–1.55)	1.07 (0.69–1.67)	1.18 (0.73–1.90)	1.31 (0.81–2.12)	1.31 (0.81–2.11)
No	ref	ref	ref	ref	ref
Car ownership					
No	**1.29 (1.00–1.69)**	1.30 (0.96–1.76)	1.31 (0.93–1.85)	1.31 (0.91–1.86)	1.25 (0.88–1.79)
Yes	ref	ref	ref	ref	ref
Smoke status					
Current smoker	1.20 (0.84–1.71)		1.04 (0.69–1.55)	1.11 (0.74–1.66)	1.13 (0.75–1.70)
Ex-smoker	**1.64 (1.07–2.51)**		**1.46 (1.01–2.39)**	**1.53 (1.02–2.53)**	**1.53 (1.02–2.52)**
Never smoker	ref		ref	ref	ref
Recreational drug use					
Current user	0.94 (0.62–1.41)		0.93 (0.57–1.51)	0.90 (0.55–1.47)	0.93 (0.57–1.52)
Ex-user	1.19 (0.89–1.60)		1.27 (0.91–1.78)	1.23 (0.88–1.72)	1.22 (0.87–1.71)
Never use	ref		ref	ref	ref
Had oral sex					
Yes	0.85 (0.62–1.16)		0.83 (0.58–1.20)	0.79 (0.55–1.15)	0.83 (0.57–1.21)
No	ref		ref	ref	ref
Self-rated general health					
Excel/very good/Good	1.31 (0.92–1.86)			1.50 (0.96–2.33)	1.48 (0.95–2.31)
Fair/Poor	ref			ref	ref
Self-rated oral health					
Excel/very good/Good	0.95 (0.71–1.25)			0.81 (0.58–1.13)	0.83 (0.59–1.61)
Fair/Poor	ref			ref	ref
Ever received HPV vaccination					
No	0.92 (0.59–1.45)				0.96 (0.57–1.61)
Do not know	0.72 (0.44–1.17)				0.78 (0.45–1.36)
Yes	ref				ref

Notes: Model 1: crude model; Model 2: adjusted for the sociodemographic factors (age, gender, geographic location, education level, income, health care card holder, car ownership); Model 3: plus adjusting for health-related behaviours (smokes and recreational drug status, and sexual behaviours); Model 4: plus adjusting for self-rated general and oral health; Model 5 (full model): plus adjusting for self-reported HPV vaccination status. Bold were donated as statistically significant differences. Bolds were donated as statistically significant differences.

## Data Availability

The datasets generated and/or analysed during the current study are not publicly available due to privacy issues of the participants. Data are available from the corresponding author on reasonable request.

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
