# Peer review of "Natural History of Oral HPV Infection among Indigenous South Australians"

_viruses, 2023, doi:10.3390/v15071573_

Round 1

Reviewer 1 Report (Previous Reviewer 2)

I would like to thank the authors for incorporating the suggested changes

Author Response

Thanks.

Reviewer 2 Report (Previous Reviewer 3)

The manuscript has been significantly improved  and it is suitable for publication in viruses

Author Response

Thanks.

Reviewer 3 Report (Previous Reviewer 4)

 This is a revision of the manuscript “Natural history of oral HPV infection among Indigenous South Australians” by Xiangqun Ju and colleagues.

The authors previously published two papers regarding this cohort (ref 29 and 30), specifically on the impact of sociodemographic factors on HPV prevalence at time 0 and 12 months, respectively. The first concern is the originality of the present investigation, considering that most cases contributing to HPV prevalence over two years were already analyzed before. Second, the discussion section lacks a comparison of the results of these three investigations. Tirth, but the more disappointing one, most of the presented data and the main message reported in the conclusion focus on static data, that is, a cross-sectional study of HPV prevalence, while the study design is instead a longitudinal study and should be more focused on dynamic data, such as incidence, persistence, clearance, and relapse

Lines 175-176: By definition, prevalence is the proportion of a population who have a specific characteristic in a given time period. There is no reason to change the standard term “prevalence” with the unusual misleading term “prevalence proportion.” As regard of the denominator in the formula, what is the meaning of “at start of the period”?

Table 1. The reviewer asked for the inclusion of counts in the tables. Including raw data is important because it allows the reader to better understand how the authors performed calculations, to check for possible inconsistencies, and to use data for meta-analyses. The authors edited the tables, including some numbers; however, missing data were not reported. Indeed they indicated n=993, but for some variable n is different, for example considering geographic location 622+396=991

Table 2. In this table, in contrast with all the others, the prevalence lacks a 95% confidence interval.

Previous comment/response 9.

Comment: Table 2. The sum of all patients with HPV type =16, 18, 31, 33, 35, 39, 45, 51, 52, 56, 59, 66, 68 and 82 does not match with the number of patients affected by HR-HPV at any of the three time points. This is a nonsense. Response: We have deleted sum of hr-HPV type (Table 2).

New comment: There is no reason to delete the data regarding hr-HPV types from Table 2. The point is that the data must be accurate. That means that the number of patients positive for hrHPV must be equal to the sum of patients affected by each HPV type =16, 18, 31, 33, 35, 39, 45, 51, 52, 56, 59, 66, 68 and 82.   A similar error is still present in the revised manuscript. Indeed, in the text at line 209 the hrHPV prevalence has been reported, but the values differ with respect to the sum of prevalence of types 16, 18…82 shown in Table 2.

Table 3. This reviewer wonders about cases that the authors defined as a fluctuation. A case -/+/- is an example of fluctuation. The reviewer believes that this case should be considered twice, both for incidence and clearance calculation. Similarly, a case +/-/+ should be considered both for incidence and clearance, or maybe relapse.

To better understand and describe the infection dynamics over the two years, or the natural history of HPV infection as defined by authors, it should be better to reconsider persistence cases when patients was infected by different HPV types at different time points.

Table 4. To calculate the prevalence stratified by sociodemographic factors, the numbers reported in Table 1 are required.  Consider the opportunity to collapse Tables 1 and 4 into a single new table.

Table 4 and 5. How did data from three different time points collapse to perform the analyses presented in Tables 4 and 5? Is it an analysis of prevalence over a two-year interval? This needs to be explained.  

Table 5. It reports the results of logistic regression. The authors stated that the study was conducted according to the STROBE guidelines. These guidelines encourage researchers to report unadjusted estimates together with those adjusted for potential confounders. The guidelines also suggest reporting criteria for excluding or including variables in the statistical models. The authors are encouraged to integrate these data.

Logistic regression usually provides an odds ratio (OR) to evaluate association levels. The authors described association results using prevalence ratio instead. Did the author manage the statistics program to produce an unusual output or did they use misleading terminology?

Author Response

This is a revision of the manuscript “Natural history of oral HPV infection among Indigenous South Australians” by Xiangqun Ju and colleagues.

  1. The authors previously published two papers regarding this cohort (ref 29 and 30), specifically on the impact of sociodemographic factors on HPV prevalence at time 0 and 12 months, respectively. The first concern is the originality of the present investigation, considering that most cases contributing to HPV prevalence over two years were already analyzed before. Second, the discussion section lacks a comparison of the results of these three investigations. Tirth, but the more disappointing one, most of the presented data and the main message reported in the conclusion focus on static data, that is, a cross-sectional study of HPV prevalence, while the study design is instead a longitudinal study and should be more focused on dynamic data, such as incidence, persistence, clearance, and relapse.
  • We have added a paragraph to compare the results of three investigations in Discussion section (Third paragraph, page 15).
  • We have added data analyses on risk factors associated with incidence, persistent and clearance, and reported in results section (Tables 5-6 and S7-8, page 11 and page 13), and discussed in discussion section (Second and last paragraphs, page 15)

  1. Lines 175-176: By definition, prevalence is the proportion of a population who have a specific characteristic in a given time period. There is no reason to change the standard term “prevalence” with the unusual misleading term “prevalence proportion.” As regard of the denominator in the formula, what is the meaning of “at start of the period”?

Thanks, we have changed to ‘period prevalence’ to distinguish from ‘point prevalence’ (ast paragraph, page 4)

  1. Table 1. The reviewer asked for the inclusion of counts in the tables. Including raw data is important because it allows the reader to better understand how the authors performed calculations, to check for possible inconsistencies, and to use data for meta-analyses. The authors edited the tables, including some numbers; however, missing data were not reported. Indeed they indicated n=993, but for some variable n is different, for example considering geographic location 622+396=991

We have reported missing data for each variable (Table 1)

  1. Table 2. In this table, in contrast with all the others, the prevalence lacks a 95% confidence interval.
  2. We have added 95% CI for prevalence (Table 2).

Previous comment/response 9.

  1. Comment: Table 2. The sum of all patients with HPV type =16, 18, 31, 33, 35, 39, 45, 51, 52, 56, 59, 66, 68 and 82 does not match with the number of patients affected by HR-HPV at any of the three time points. This is a nonsense. Response: We have deleted sum of hr-HPV type (Table 2). New comment: There is no reason to delete the data regarding hr-HPV types from Table 2. The point is that the data must be accurate. That means that the number of patients positive for hrHPV must be equal to the sum of patients affected by each HPV type =16, 18, 31, 33, 35, 39, 45, 51, 52, 56, 59, 66, 68 and 82.   A similar error is still present in the revised manuscript. Indeed, in the text at line 209 the hrHPV prevalence has been reported, but the values differ with respect to the sum of prevalence of types 16, 18…82 shown in Table 2.

Thanks, we have added 4 hr-HPV type (HPV 26, 53, 58 and 73) and recalculated total number and prevalence (Table 2, second paragraph, page ;7)

  1. Table 3. This reviewer wonders about cases that the authors defined as a fluctuation. A case -/+/- is an example of fluctuation. The reviewer believes that this case should be considered twice, both for incidence and clearance calculation. Similarly, a case +/-/+ should be considered both for incidence and clearance, or maybe relapse.

Thanks. We have re-calculated incidence and clearance which included two fluctuation scenarios (Table 3), and re-defined incidence and clearance, reported results and discussed in Method, Result and Discussion sections, as well as in the Abstract.

  1. To better understand and describe the infection dynamics over the two years, or the natural history of HPV infection as defined by authors, it should be better to reconsider persistence cases when patients was infected by different HPV types at different time points.

We have added a Table to report persistence cases with different HPV types at different time points (Table S3), and reported in text (Second paragraph, page 7).

  1. Table 4. To calculate the prevalence stratified by sociodemographic factors, the numbers reported in Table 1 are required.  Consider the opportunity to collapse Tables 1 and 4 into a single new table.

We have combined Table 1 and Table 4 into a new table (Table 1) and delated Table 4.

  1. Table 4 and 5. How did data from three different time points collapse to perform the analyses presented in Tables 4 and 5? Is it an analysis of prevalence over a two-year interval? This needs to be explained.  

Yes, it was an analysis of prevalence over a two-year interval. We have re-named prevalence as period prevalence as advised and explained in Methods section (Lase paragraph, page 4),

  1. Table 5. It reports the results of logistic regression. The authors stated that the study was conducted according to the STROBE guidelines. These guidelines encourage researchers to report unadjusted estimates together with those adjusted for potential confounders. The guidelines also suggest reporting criteria for excluding or including variables in the statistical models. The authors are encouraged to integrate these data.

We have added the details on statistical models (Second paragraph, page 5), created Tables to report unadjusted and adjusted estimated results together (New Table 4, 6 and S4 to S8), and reported results (Last paragraph, page 10, Second paragraph, page 13) and discussed our findings (Second paragraph, page 15).

  1. Logistic regression usually provides an odds ratio (OR) to evaluate association levels. The authors described association results using prevalence ratio instead. Did the author manage the statistics program to produce an unusual output or did they use misleading terminology?

No, we did not estimate OR, the following list provides explanation (see reference):

  • Due to the high prevalence of the outcome (the prevalence of any oral HPV infection was 51.3%), we chose PR over OR, as OR would have “overestimated” the strength of the association considerably.
  • When confounding is defined using “collapsibility”, RR and not the OR is an intrinsic measure of interest. Our ‘Sexual behaviour’ was ‘collapsibility’ variable.
  • When the proportion of outcome is “rare” (e.g <10%) OR and PR are closer to each other (the prevalence of HPV16/18 infection was 4.5%).

Therefore, we have chosen to estimate PR not OR.

Reference:

Tamhane AR et al. Prevalence Odds Ratio versus Prevalence Ratio: Choice Comes with Consequences. Stat Med. 2016; 35(30): 5730:5735. Doi: 10.1002/sim.7059.

Round 2

Reviewer 3 Report (Previous Reviewer 4)

The manuscript was significantly improved.

This manuscript is a resubmission of an earlier submission. The following is a list of the peer review reports and author responses from that submission.

Round 1

Reviewer 1 Report

The results of this study represent the first published results on oral HPV infection among the population of South Australia, and therefore I consider the work important and very interesting.

But I think that the presentation of the results - statistical analysis is too extensive and therefore confusing, especially in Table 5.

When the authors change the statistical presentation and emphasize only what is statistically significant, it will be easier for the reader to follow what the authors wanted to show and what is important.

Reviewer 2 Report

Review viruses-2322425, Ju et al., Natural history of oral HPV infection among Indigenous South Australians

The authors describe the prevalence, incidence, persistence and clearance of oral HPV infections in indigenous Australian adults, based on three time points over a 2-year period. This is a continuation of a study of which the baseline and 1-year follow up have been published previously (references 24 and 25). As such, the baseline prevalence and some of the incidence and clearance results have already been published. Furthermore, lines 217-218 are somewhat misleading, as ‘only’ the 2-yr follow up is a first. The authors should mention the two earlier papers in the introduction.

Line 140: It is not clear from Figure 1 where N=993 comes from. N= 993 also doesn’t seem to match N=910 as indicated in Table 2. Please, explain in the text or Figure.

Lines 164-166: Could the authors comment in the discussion on the apparent decrease in HR-HPV and HPV16/18 prevalence from 0 to 24 months? Could this be due to changes in sexual behaviours during the COVID-19 pandemic?

Lines 174-176: It is unclear if persistence means that the same type(s) of HPV was detected at all three time points. Does persistence mean that the same HPV type was present at 0,12 and 24 months? Or could it also be that one HPV type was present at 0 and 12 months and then another HPV type at 24 months? In case of the latter, where the HPV types can be different between time points, the analyses should be repeated to generate results on persistence of the same HPV type.

Table 3 – The authors should explain why a patterns of x üx and ü x ü are referred to as fluctuations? Why should x üx not be considered as an incident infection with a clearance within 1 year? That assumption was previously made by the authors of this manuscript in Sethi et al, Cancer Epidemiol Biomarkers Prev 2022, where for the 0,12 months analysis an outcome of üx was considered a clearance. As another example, Antonsson et al. (Int J Cancer 2021) define oral HPV clearance as a sample returning a negative HPV test for the same HPV type from a participant who had previously returned a HPV-positive sample.

Similarly, ü x ü could be considered a prevalent infection with clearance and a new infection, especially if the HPV type at 24 months is different from the type at baseline. Please, explain why this defined as a fluctuation.

Line 264 – related to the comment on lines 164-166 above, do the authors think that the COVID-19 pandemic may have influenced the outcomes of the study? If so, this should be mentioned as a (unfortunate) limitation.

Minor comments:
Line 37: ‘infection’ instead of ‘infectious’

Line 176: ‘individual)’ instead of ‘individua)’

Lines 254-255: Suggestion to say: ‘45 out of 54 (=83%) cases of HR-HPV were due to HPV16/18.’

Reviewer 3 Report

In the manuscript entitled “Natural history of oral HPV infection among Indigenous South Australians” the authors present significant data concerning the prevalence of oral HPV infection among Indigenous south Australians. The manuscript is very informative and well-written. However, some points need to be addressed.

·   The introduction must be improved. In particular, authors are required to provide more details concerning the natural history of HPV infection. Τhe description of viral DNA structure would help the readers to better understand the biology of HPVs. Moreover, it would be beneficial to mention more information concerning the phylogenetic classification of HPV genome as well as to describe in detail the members of HR and LR HPV genotypes, including the criteria that were used to classify the HPV genotypes into these two groups. (Front. Oncol. 2019;9:355. doi: 10.3389/fonc.2019.00355, Expert Rev Mol Med. 2021;23:e19. doi:10.1017/erm.2021.18, Virol. J. 2010;7:11.doi: 10.1186/1743-422X-7-11).

·    The tumorigenic mechanisms that high-risk HPV genotypes use to promote malignancy development are required to be reported.

·   There are no epidemiological findings concerning the global incidence and mortality of oropharengeal cancer.

·   There is no information concernig the HPV vaccination status of the examined individuals.  Were they all unvaccinated? 

·  In Materials and Methods the methodologies that were used for DNA extraction, HPV detection and genotyping are not reported in detail. Moreover, it is not mentioned whether HPV positive DNA samples were subjected to sequencing to confirm viral DNA sequences.

·    The present study focuses on oral HPV infection, thus providing important data concerning the prevalence of various HPV genotypes in the examined individuals. I would like to ask the authors whether HPV DNA test could be considered as a choice screening method for oropharyngeal cancer prevention in the future.

Reviewer 4 Report

The manuscript “Natural history of oral HPV infection among Indigenous South 2 Australians” by Xiangqun Ju and colleagues reports a longitudinal study of HPV infections among Indigenous populations of Australia. Overall the investigation is interesting and well-planned, however, the manuscript needs to be improved, mainly in method and in data presentation. It follows some suggestions.

Lines 79-81 and Fig.1 Number of patients lost to follow-up was much higher than 9 death, 52 travel interstate, and 1 incarceration, as reported in fig.1. Please add more withdrawal reasons or include a simple “other”.

Fig. 1 lacks of caption and was not mentioned in text.

Lines 104-111. The paragraph needs to be revised by integrating more details. A not exhaustive example of information needed regards the method of DNA purification, reagents and hardware used for PCR, references for the used primers, methods for revelation of PCR products and so on. Authors need to add the methods used for HPV type identification.

Line 121-127 This part is hard to figure out and sometimes wrong. Easier to understand when looking at Table 3. The sentence “Incidence was defined as negative sample at baseline and/or at 12 months, followed by a positive sample at 24 months;” has a different meaning from what is seen in table 3.

Lines 127-129 This sentence do not add anything to the sentence at line 120 “among 120 those who had valid (β-globin positive) samples at all three timepoints”

Line 129-131 This reviewer does not get the meaning of this sentence.

Line 146 and Table 1 caption wrongly report “HPV infection”. This data includes all samples with affected and unaffected patients.

Line 147 includes a wrong statement due to the above.

Line 162-166 Consider revising this sentence, prevalence is not a race.

Table 2. The sum of all patients with HPV type =16, 18, 31, 33, 35, 39, 45, 51, 52, 56, 59, 66, 68 and 82 does not match with the number of patients affected by HR-HPV at any of the three time points. This is a nonsense.

Table 3. The table should include counts, not only percentages.

Table 4. The table should include counts, not only percentages, as well as unaffected counts. The caption state “Incidence proportion”, if I am not wrong, the table shows “prevalences” instead. In case, revise also the description in the text.